behaviour, cognition, evolution

punishment, signalling, cooperation, partner choice, trust, reputation

**Author for correspondence:**
Nichola J. Raihani
e-mail: nicholaraihani@gmail.com

# Third-party punishers do not compete to be chosen as partners in an experimental game

Tommaso Batistoni[1], Pat Barclay[2] and Nichola J. Raihani[3]

[1]Centre for Experimental Social Sciences, University of Oxford, Oxford OX1 1NF, UK
[2]Department of Psychology, University of Guelph, Guelph, Canada N1G 2W1
[3]Department of Experimental Psychology, University College London, London WC1H 0AP, UK

PB, 0000-0002-7905-9069; NJR, 0000-0003-2339-9889

Third-party punishment is thought to act as an honest signal of cooperative intent and such signals might escalate when competing to be chosen as a partner. Here, we investigate whether partner choice competition prompts escalating investment in third-party punishment. We also consider the case of signalling via helpful acts to provide a direct test of the relative strength of the two types of signals. Individuals invested more in third-party helping than third-party punishment and invested more in both signals when observed compared to when investments would be unseen. We found no clear effect of partner choice (over and above mere observation) on investments in either punishment or helping. Third-parties who invested more than a partner were preferentially chosen for a subsequent Trust Game although the preference to interact with the higher investor was more pronounced in the help than in the punishment condition. Third-parties who invested more were entrusted with more money and investments in third-party punishment or helping reliably signalled trustworthiness. Individuals who did not invest in third-party helping were more likely to be untrustworthy than those who did not invest in third-party punishment. This supports the conception of punishment as a more ambiguous signal of cooperative intent compared to help.

## 1. Introduction

Punishment refers to the act of paying a cost to inflict a reciprocal cost on a social partner [1] and it has been proposed as a key factor supporting the evolution of cooperation among non-relatives [2–7]. Individuals who punish defectors can generate collective benefits if punishment increases within-group cooperation but punishment also imposes an individual cost on the punisher. As such, to understand how punishment can be under positive selection, one must ask how punishers might benefit from making these costly investments. One route to obtaining return benefits from punishment is if the target of punishment behaves more cooperatively in future interactions with the punisher (as originally suggested by [1]). However, such outcomes seldom seem to occur in experimental settings (reviewed in [7]). Moreover, both laboratory (e.g. [8]) and field experiments (e.g. [9,10]) have shown that people often punish in situations where they act as third-parties, who were not the primary victim of the cheat and do not expect to interact with either the victim or the cheater in future interactions.

An alternative route by which punitive strategies could yield individual-level benefits to punishers is via reputation consequences that increase the punisher's likelihood to have profitable social interactions in the future [11–18]. Building a reputation as a punisher might yield benefits in two distinct ways [16]: (i) by signalling formidability, which can deter current social partners or bystanders from cheating when they interact with individuals with a punitive reputation (e.g.

[12,19]); or (ii) by signalling cooperative intent, such that punishers benefit from increased access to cooperative interactions with new social partners (e.g. [11,15,17,20,21]). Here we focus on the latter possibility.

Cooperation may act as a signal, allowing cooperators to encourage existing partners to cooperate and/or to attract high-quality social partners (or partners who are committed to cooperating, [22–24]) for interactions. The potential to attract partners becomes particularly salient when individuals are embedded in fluid social networks and are therefore able to break and form social ties. In fluid social networks, partner choice can introduce a market-like logic, resulting in an increased level of competition among individuals to be chosen by the best partners [25–31]. Confirming this perspective, previous work has shown that in so-called biological markets [28] cooperation levels can escalate [32] and are often higher compared to when there is no possibility for partner choice (e.g. [23,27,33]; for recent reviews, see [25,34]).

Punitive acts can also be conceptualized as signals that allow the punisher to convey an otherwise unobservable intent to cooperate [15,35,36]. As long as the production of the signal (i.e. the punitive act) is associated with the hidden quality (i.e. cooperativeness), observers can then act contingently on the informative value of the signal; and may be more likely to reward punishers for their actions [17], to trust punishers [11,15], or to select punishers (over non-punishers) as partners [16].

Nevertheless, the motives underpinning punishment decisions are hard to discern because punishment is, by definition, a harmful act. Punitive strategies could stem from spiteful motives, aimed at harming other people, or from fairness concerns and desires to uphold social norms of behaviour [7,20]. Punishment is, by definition, a more ambiguous signal of cooperative intent than is helping, since helping others is less likely to stem from harmful or competitive motives. Indeed, recent evidence shows that people who invest in third-party punishment (rather than third-party compensation) are more likely to score highly for antisocial personality traits, such as Machiavellianism, narcissism and psychopathy [37], suggesting that third-party punishment is less likely to be an honest signal of cooperative intent.

The information punishment conveys—and its associated reputation consequences—are likely to be highly context-specific [16]. For example, when individuals are given an option to compensate a victim or to punish a cheat, then punishment is less likely to signal cooperative intent [15,17,37,38]. Contexts where the punisher is harmed directly by the wrongdoer (second-party punishment) are more likely to be motivated by vengeful sentiments and less likely to be interpreted as signals of the punisher's cooperativeness [16,38]. Conversely, when the punisher is an uninvolved bystander to the initial 'crime' (third-party punishment), then punitive acts are more likely to convey cooperative intent. Punishment that is cheaper to administer than the damage it inflicts on a target could also be perceived as a competitive act [7,16], rather than signalling an intent to cooperate. The potential ambiguity and context-dependency of punishment means that, especially in decontextualized settings typical of economic games, the reputation consequences of punishing others might not always be positive [16,36–40]. In some cases, therefore, individuals might choose to hide investments in punishing others [41] or preferentially invest in helping rather than punishing others when these decisions will be revealed to others [42]. Conversely, because helpful acts entail paying an individual cost to generate benefits to others, the motives underpinning these acts are less likely to be affected by context and may be more likely to be perceived as stemming from genuinely helpful motives (but see [43]).

A simple hypothesis that does not take ambiguity and context-specificity into account might posit that if punishers stand to gain reputation benefits from punishing, then they should invest more in punishment when these decisions will be made known to other individuals. Similarly, we might expect punishers to further escalate their investments when there is competition to be selected as a social partner, as it has been previously shown for helping behaviour [44]. This hypothesis might, however, be unsupported if punishment is an ambiguous signal of cooperative intent and if increasing investments in punishment cast further doubt on the punisher's underlying intentions and commitment to cooperation. Specifically, because the motives underpinning punishment may not indicate a cooperative disposition, we might not expect people to form favourable impressions of those who make escalated investments in punishment. As a consequence, people may not reward or trust or prefer to interact with people who invest high amounts in punishment, even if they do preferentially reward, trust and prefer to interact with people who invest high amounts in helpful behaviour.

In this study, therefore, we asked whether punishers do escalate their investments in punishment when there is the potential to be chosen by an observer—and whether punishers are preferred as interaction partners and entrusted with more money. We also examine whether investments in punishment act as honest signals of cooperative intent, by measuring whether punitive investments are related to trustworthy behaviour in a game where individuals are financially incentivized to exploit a partner. Importantly, we compare whether investments in punishment escalate to the same extent as investments in helping behaviour—and whether these signals are treated differently by potential interaction partners. This part of the study aims both to replicate previous results exploring competitive signalling of generosity [27] as well as to explore any differences in how people use and respond to signals of punishment versus signals of helping behaviour.

We aimed to test the hypothesis that punishment might be used as a signal of cooperative intent under the most conducive conditions, by using an experimental setting that reduced the potential ambiguities discussed above. As such, we explored the potential signalling value of punishment in a third-party punishment paradigm, where there is a stronger theoretical argument for punishment to operate as a signal of cooperative intent [16,38]. Furthermore, we used a fee-to-fine ratio of 1 : 1, meaning that punishers could not use punishment to elevate their own pay-offs relative to those of the target. This should reduce the possibility to infer that punishment is driven by spiteful preferences rather than an impartial concern for fairness [7,16].

The prospect of positive reputation gains may increase investments in punishment (or helping) either because the third-party knows that their past behaviour will be revealed to others (who may treat them differently on the basis of previous actions) or because the third-party anticipates gaining access to social partners on the basis of their investment. To distinguish between these two possibilities, we replicate Barclay & Willer's [27] design by including a 'knowledge only' condition, in addition to a 'knowledge + partner choice'

condition. This allows us to test whether any increased investment relative to baseline stems from the fact that one's behaviour will be advertised to a potential social partner and whether investments escalate even further when individuals can compete to be chosen as a partner for future social interactions. If third-parties' investments are used to compete to access new social partners, then we expect them to be higher in the knowledge + partner choice condition than they are in the knowledge only condition.

To summarize, in the current study we tested the following hypotheses: H1 to H3 concern third-party signalling; H4-H6 concern whether bystanders selected third-parties on the basis of these signals; and H7 and H8 deal with the extent to which third-party investments predict those individuals' trustworthiness.

Third-party signalling:

— H1: observability increases investments in both (i) third-party punishment and (ii) helpful behaviour relative to a baseline where no observation is possible;
— H2: the possibility for partner choice leads to higher investments in (i) third-party punishment and (ii) costly helpful actions compared to the mere observability of those actions by bystanders; and
— H3: third-parties prefer to invest in helping (rather than punishment) when signalling to potential interaction partners.

Bystander choices:

— H4: investments in (i) third-party punishment and (ii) help positively predict probability to be chosen as a social partner by bystanders;
— H5: third-parties who invest more in (i) punishment and (ii) help are trusted more by bystanders; and
— H6: bystanders place more weight on signals of help than on signals of punishment when (i) choosing and (ii) trusting a partner for a cooperative interaction.

Reliability of third-party signals:

— H7: third-party investments in (i) punishment and (ii) help are positively associated with trustworthiness; and
— H8: investments in third-party help more reliably predict trustworthiness than investments in third-party punishment.

## 2. Methods

### (a) Participants

All data, code and materials to reproduce this study, including instructions shown to participants, can be found at https://osf.io/4zpkb/. This study was approved by the UCL Research Ethics Committee (project 3720/001). All data were collected in 2018 and participation was voluntary. We recruited 2253 participants through the online labour market Amazon Mechanical Turk (MTurk). Each participant was allocated to one of three roles: dictator ($n = 902$), third-party ($n = 902$) and bystander ($n = 449$). Throughout the study, roles were labelled using neutral terms. All participants received a show-up fee contingent on the role assigned ($0.20 for dictator, $0.50 for both third-party and bystander) and were given the chance to earn a bonus based on their decisions in the experiment. Total average earnings for each role were $0.52 (dictators), $1.12 (third-parties) and $0.80 (bystanders).

All data were collected anonymously and no deception was used. The predictions in this study were not pre-registered.

### (b) Experimental design

Third-parties and bystanders were assigned to one of six treatments (described below). After reading the instructions, third-parties and bystanders completed a comprehension check (comprising eight questions) and then made decisions in the experiment. Of the third-parties, 51.7% participants answered all comprehension questions correctly; whereas 51.4% of bystanders answered all comprehension questions correctly. All data are included in the main analysis to avoid selection bias, although we include comprehension as an explanatory term in our models. We also re-ran the main models excluding participants who failed one or more comprehension checks and report qualitative differences.

The experimental setting consisted of three stages. In stage A, dictators and third-parties played a variant of the Dictator Game [45]. Each dictator was endowed with $0.50 and faced a dichotomous decision between a fair (keep $0.25 and give $0.25) and an unfair (keep $0.45 and give $0.05) share of the endowment with a passive receiver. Receivers were unrecruited MTurk workers who had taken part in a previous study run in our laboratory, who received a bonus according to the decision made by their matched dictator partner. Following the dictator decision, the third-party chose how much of their endowment, if any, to invest to punish an unfair dictator or, according to the experimental condition, to help a receiver who was given an unfair share. Third-parties were endowed with $0.50 and could invest any amount (in $0.01 increments) between $0.00 and $0.45 to either help or punish the target individual, with a fee to fine/fee to help ratio of 1:1. Importantly, allocation to help and punishment conditions was a between-subjects variable, so third-parties could either choose to punish (or not punish) or they could choose to help (or not help) but they did not choose between punishment and helping. Third-party decisions were made using the strategy method and third-parties were informed that their decision would be implemented if they were matched to an unfair dictator. The strategy method involves players making conditional decisions for all potential scenarios in an economic game and is widely used in behavioural economics, where it is thought to produce reliable results [46].

In stage B, third-parties were randomly paired with another third-party, and one in each pair was selected (either randomly or actively chosen by a bystander, according to the experimental condition) to take part in stage C.

In stage C, the selected third-party and a bystander played a Trust Game [47] as the trustee and the trustor, respectively. The bystander was given $0.30 and could choose how much, if any, to send to the third-party. The amount sent was tripled and the third-party could choose what percentage (0–100% in 10% increments) to keep for themselves and what percentage to return to the bystander; the amount returned provides a measure of trustworthiness.

We implemented a 2 (punish versus help) × 3 (random allocation/no knowledge; random allocation/knowledge; partner choice/knowledge) between-subjects fractional factorial design, resulting in six experimental treatments. Across treatments, we varied: (i) whether third-parties could punish an unfair dictator versus help the corresponding receiver in stage A; (ii) whether third-parties were randomly allocated to versus actively chosen by a bystander in stage B; and (iii) whether the bystander was informed versus not informed of third-parties' behaviour in stage A. Third-parties received different instructions about the potential consequences of their helping or punishment investment in game B, according to the condition they were allocated to. Specifically, workers who were assigned to the random allocation condition saw text stating, 'Game B. To take part in this game, you will have to be RANDOMLY selected to interact with a NEW

worker, who will be Player 4'; whereas workers who were assigned to the partner choice condition saw text stating, 'Game B. To take part in this game, you will have to be CHOSEN by a NEW worker, who will be Player 4'. Third-parties who were assigned to the no knowledge condition also saw text stating, 'Player 4 will NOT participate in GAME A and will NOT know how you behaved in GAME A'. Third-parties allocated to the knowledge condition saw text stating, 'Player 4 will NOT participate in GAME A but WILL know how you behaved in GAME A'.

Allocations to all treatments occurred randomly within each session and participants made their decision in isolation. Participants were matched to a partner using ex-post matching [48]. Across treatments, four third-parties could not be paired with another third-party and were therefore not matched with a bystander (but since decisions were collected using the strategy method, these are retained in the analysis). Details on the procedure and the matching protocol, as well as experimental instructions and comprehension questions are provided in the SI.

## (c) Analysis

All data were analysed using R (version 1.4.1717). Below we report how we tackled each hypothesis, in turn. We present estimates from a generalized linear model (GLM) output, which can be understood as regression coefficients. All data and code to reproduce models are available at https://osf.io/4zpkb/.

> H1: observability increases investments in both (i) third-party punishment and (ii) helpful behaviour relative to a baseline where no observation is possible.

> H2: the possibility for partner choice leads to higher investments in (i) third-party punishment and (ii) costly helpful actions compared to the mere observability of those actions by bystanders.

> H3: third-parties prefer to invest in helping (rather than punishment) when signalling to potential interaction partners.

To test whether third-parties invested more in punishment (or helping) when their decisions would be revealed to a future partner (H1) or when they might be chosen on the basis of their decision (H2), third-party investments were fitted as the response term to a GLM, with treatment (anonymous/random, knowledge/random, knowledge/choice), condition (punishment/helping) and comprehension (1/0, determining whether the participant correctly answered all the relevant comprehension questions) included as explanatory terms. This also allowed us to test whether third-parties preferentially invested in helping over punishment, and whether this preference varied according to treatment (anonymous/random, knowledge/random, knowledge/choice) (H3). We also included the following two-way interactions: condition × treatment (to test whether there was a difference between tendency to signal in the helping versus the punishment condition) and treatment × comprehension (to account for the possibility that failed comprehension might impact investment more in some treatment conditions than in others). 'Anonymous/random' was set as the reference category in treatment and we subsequently re-ordered the levels of this factor to test for differences between knowledge/random and knowledge/choice conditions.

> H4: investments in (i) third-party punishment and (ii) help positively predict probability to be chosen as a social partner by bystanders.

> H5: third-parties who invest more in (i) punishment and (ii) help are trusted more by bystanders.

> H6: bystanders place more weight on signals of help than on signals of punishment when (i) choosing and (ii) trusting a partner for a cooperative interaction.

We addressed H4 by performing two binomial tests asking whether the higher investor was more likely to be chosen in the help and punishment conditions, respectively. To address H6(i),

we also ran a chi-squared test to explore whether the probability of the higher investor being chosen varied according to condition (punishment/help). Because this approach meant that we could not use data from the 32 pairs (22 in punishment condition, 10 in help condition) where both third-parties invested the same amount, we also ran an additional model that allowed us to use all the data, including cases where both third-parties invested the same amount. Chosen (1/0) was set as the binomial response term in a logistic regression with the terms 'condition' (help/punishment) and 'highest investor' (an ordered categorical variable with three levels, where lower < equal < higher denoting whether the third-party was the lower, equal or higher investor of the pair) set as explanatory terms. We also included the two-way interaction between these terms to estimate whether being the highest or lowest investor differentially impact the probability to be chosen in the help rather than the punishment condition.

To test H5 and H6(ii), we set the amount entrusted to the third-party as the dependent variable in a GLM and included the following explanatory terms: invest (the amount the third-party invested in punishment/helping), condition (1/0, denoting whether the condition was punishment or help) and the two-way interaction between these terms.

> H7: third-party investments in (i) punishment and (ii) help are positively associated with trustworthiness.

> H8: investments in third-party help more reliably predict trustworthiness than investments in third-party punishment.

To test H7 and H8, we set the percentage of the endowment returned to the truster as the dependent variable in a GLM, including the following explanatory terms: invest (the amount the third-party invested in punishment/helping), condition (1/0, denoting whether the condition was punishment or help) and the two-way interaction between these terms

## 3. Results

Most participants (255 out of 443, 57.6%) did not punish the selfish dictator. The mean (± s.e.) amount invested in punishment was $0.09 ± 0.01, with players investing anything from $0.00 to $0.45 to punish a third-party. Investing nothing was also the most common decision in the help condition, with 131 out of 459 (28.5%) individuals investing nothing to help a receiver (range: $0.00–$0.45). However, the mean amount invested in help ($0.16 ± 0.01) was higher than mean amount invested in punishment (Wilcoxon test, W = 68998, $p < 0.001$).

> H1: observability increases investments in both (i) third-party punishment and (ii) helpful behaviour relative to a baseline where no observation is possible.

> H2: the possibility for partner choice leads to higher investments in (i) third-party punishment and (ii) costly helpful actions compared to the mere observability of those actions by bystanders.

> H3: third-parties prefer to invest in helping (rather than punishment) when signalling to potential interaction partners.

In both conditions (punishment/help), third-parties invested more both when their behaviour would be observed (estimate: 0.05, $p = 0.008$) and when there was the possibility for partner choice (estimate: 0.08, $p < 0.001$) relative to baseline (figure 1). Investments were higher in the knowledge/choice compared with the knowledge/random treatment, but this difference was not significant (estimate: 0.03, $p = 0.19$), indicating that people did not invest more when there was the possibility to be chosen as a partner, compared to when their investment was observable but did not inform a partner choice decision. Therefore, H1 was supported but H2 was not, either for helping or for punishment. Investments in the knowledge/choice

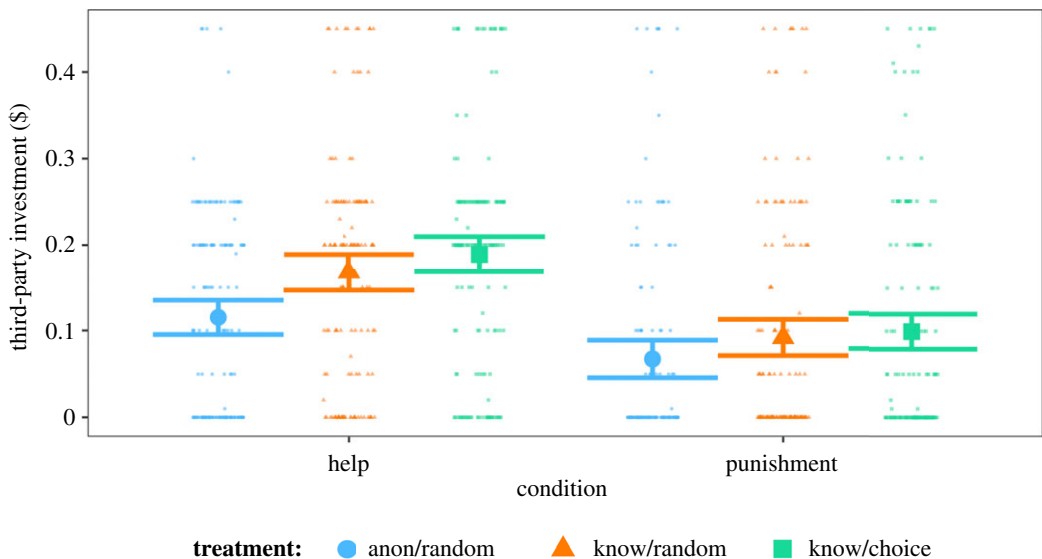

**Figure 1.** Predicted means with 95% confidence intervals from GLM exploring third-party investments in help and punishment according to treatment. Raw data showing third-party investments ($) are overlaid with a horizontal jitter of 0.3 points applied to aid visualization where several datapoints overlap. (Online version in colour.)

treatment (relative to baseline) varied depending on whether the third-party was investing in help or punishment: relative to baseline, third-parties increased their investments in punishment less than they did in help (estimate: −0.04, $p = 0.048$; figure 1), providing some support for H3. We note that this interaction effect is only marginally significant and inference should therefore be drawn with caution. We observed high levels of failed comprehension in the task: 292 out of 902 (32.4%) participants failed at least one of the five comprehension questions concerning the incentives surrounding third-party investments. Nevertheless, comprehension was not significantly associated with third-party investments in our model (estimate: −0.01, $p = 0.47$) and did not differentially affect investments across different treatments (both $p > 0.5$). We also re-ran the analyses above excluding the 292 participants who failed at least one comprehension check. This did not substantively change our findings (results presented in the electronic supplementary material).

> H4: investments in (i) third-party punishment and (ii) help positively predict probability to be chosen as a social partner by bystanders.

> H5: third-parties who invest more in (i) punishment and (ii) help are trusted more by bystanders.

> H6: bystanders place more weight on signals of help than on signals of punishment when (i) choosing and (ii) trusting a partner for a cooperative interaction.

In the punishment condition, and in those groups where the two third-parties had invested different amounts, the highest investing player was chosen as the partner on 36 out of 56 (64.3%) occasions, indicating that bystanders used this information to select social partners (binomial test, $p = 0.04$). In the help condition, and in those groups where the two third-parties had invested different amounts, the most helpful individual was chosen on 63 out of 65 occasions (96.9%; binomial test, $p < 0.001$). The higher investor was more likely to be chosen in the help condition than in the punishment condition ($\chi^2$-test, $x^2 = 19.4$, $p < 0.001$). Therefore H4 and H6(i) were both supported.

In the full model, including data from trials where both third-parties invested the same amount, the results were

similar. Specifically, the effect of the lower or higher investor of the pair being chosen varied with condition. Lowest investors were less likely to be chosen in the help condition compared to the punishment condition (estimate: 3.68, $p < 0.001$; figure 2); and highest investors were more strongly preferred in the help condition than in the punishment condition (estimate: −2.88, $p < 0.001$; figure 2).

Bystanders entrusted on average $0.18 ± 0.01 (out of $0.30) of their endowment to third-parties. Third-parties who invested more to help or punish were entrusted with more money by bystanders (estimate: 0.29, $p = 0.01$; figure 3), supporting H5. Nevertheless, bystanders did not entrust more to individuals that they chose to interact with compared with those they were randomly allocated to interact with (estimate: 0.02, $p = 0.33$) or to helpers over punishers (estimate: −0.02, $p = 0.29$). We found no evidence for two-way interactions between the amount the third-party invested and either (i) being chosen by the bystander (estimate: −0.12, $p = 0.20$) or (ii) whether the third-party was a punisher or a helper on the amount entrusted by bystanders (estimate: 0.08, $p = 0.44$). Therefore, H6(ii) was not supported. Despite high levels of failed comprehension (148 out of 298 bystanders failed at least one comprehension check), there was no significant effect of task comprehension on trust decisions (estimate: 0.01, $p = 0.43$). Excluding players who failed a comprehension check yields a stronger effect of investment on trust (estimate: 0.43, $p < 0.001$) and also yields a marginal positive effect of choice, indicating that bystanders entrusted more money to partners they had chosen to interact with (estimate: 0.06, $p = 0.047$).

> H7: third-party investments in (i) punishment and (ii) help are positively associated with trustworthiness.

> H8: investments in third-party help more reliably predict trustworthiness than investments in third-party punishment.

Third-parties returned a mean of 33.4 (±1.08) % of the endowment to bystanders. Third-parties who invested more in helping or punishing in stage B were more trustworthy (estimate: 0.87, $p < 0.001$) although we detected an interaction between investment and condition, whereby low investment in third-party help more strongly predicted untrustworthiness, compared to low investment in third-party punishment

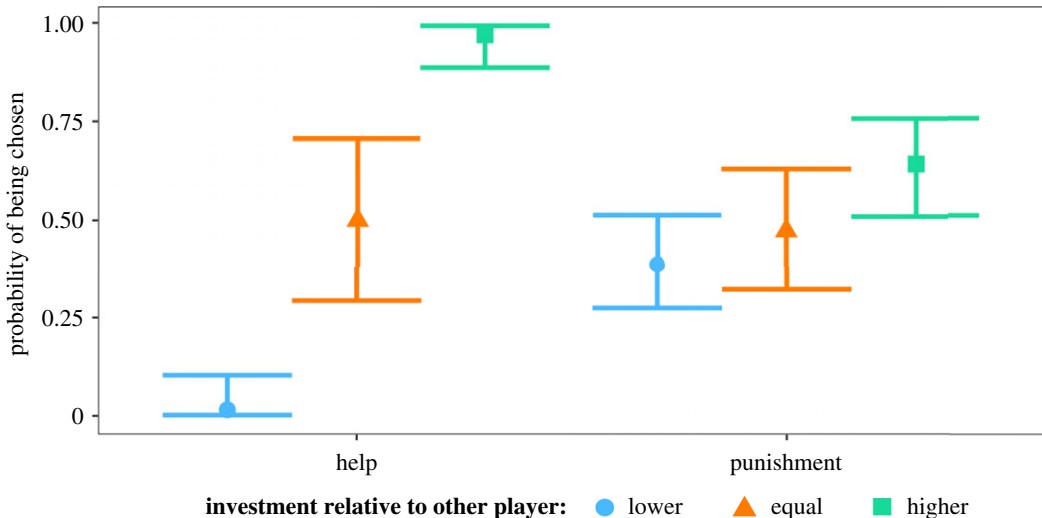

**Figure 2.** Predicted means (with 95% confidence intervals) of probability to be chosen by a bystander, as a function of whether the third-party was the lower or higher investor of the pair, or whether both third-parties invested equally. (Online version in colour.)

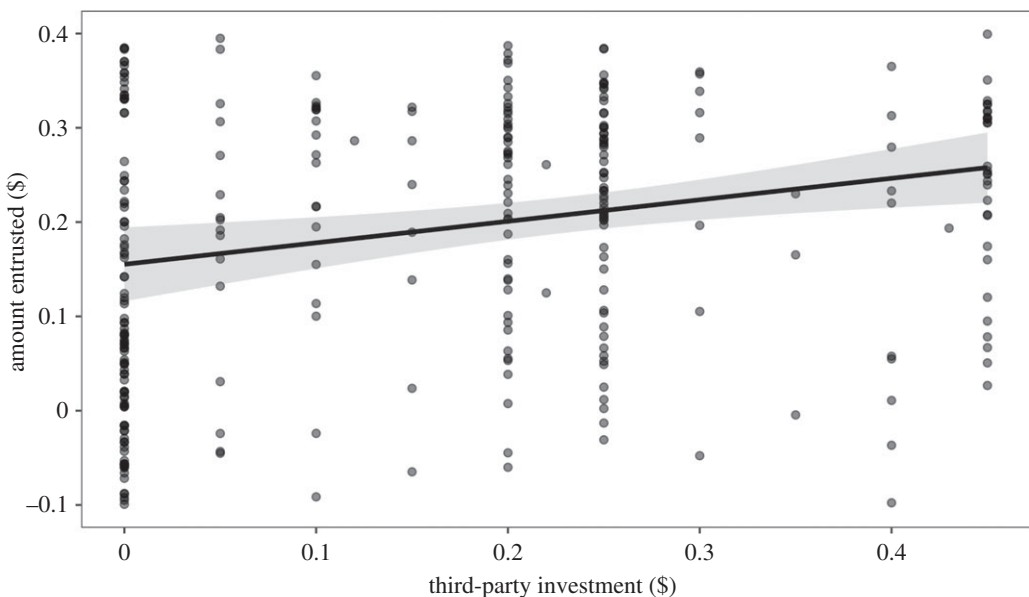

**Figure 3.** Predicted mean amount entrusted to third-parties by bystanders as a function of how much the third-party invested in help or punishment. The 95% confidence interval around the predicted mean is shown. Raw data points are overlaid with a vertical jitter of 0.1 to aid visualization, hence some points appear below zero on the *y*-axis.

(estimate: −0.47, *p* = 0.001; figure 4). Most trustees (403 out of 449) understood that they could maximize their pay-offs by not sending any money back to the truster. Re-running this model excluding the 46 participants who failed that comprehension question did not qualitatively change results (see the electronic supplementary material). We conclude that H7 and H8 were both supported.

## 4. Discussion

This study aimed to address a gap in the current literature by investigating whether investments in costly punishment are used as signals of trustworthiness and whether such signals escalate in the presence of competition to be chosen as a partner. We were also interested in the signalling value of punishment compared to helping behaviour: specifically, whether investments in punishment are less prone to escalate (compared to investments in help) when there is competition to be chosen as a partner; and whether punishment investments are viewed by observers as less reliable indicators of an individual's cooperative intent. Although we found that investments in both punishment and helping behaviour were higher when these would be observed by another individual, this effect was stronger for helping than for punishment and individuals generally invested more in helpful than in punitive behaviour. Moreover, the preference to interact with helpful third-parties was more pronounced than the preference to interact with punitive third-parties. Taken together, these results support the idea that punishment signals may be used to signal cooperative intent to a potential partner—but that these signals are more ambiguous than signals sent via investment in helping behaviour. Accordingly, individuals are less prone to invest in punishment (compared to helping) when attempting to signal their cooperative intent to others. More generally, while our results support the idea that third-party punishment could be supported by reputation benefits to punishers, there seems to be less scope for individuals to accrue a

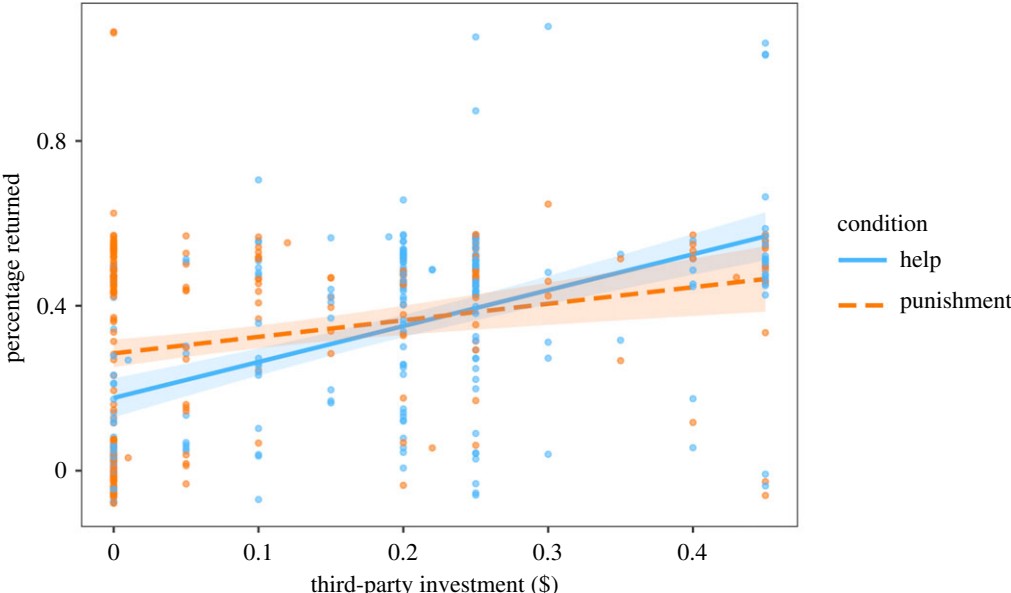

**Figure 4.** Trustworthiness varied as a function of third-party investment in punishment versus helping. Predicted means of increasing third-party investment on trustworthiness (percentage of endowment returned to truster) with 95% confidence intervals. Raw data points are overlaid with a vertical jitter of 0.05 applied to ease visualization. Investments in third-party helping and third-party punishment are both associated with increased trustworthiness. However, low investment in helping indicates low trustworthiness, but the same is not true for low investments in punishment. (Online version in colour.)

positive reputation by punishing others than to accrue a positive reputation by helping others. This would explain the sometimes contradictory findings about punishment, whereby punishers are trusted (e.g. [11,15]) but are not always liked or rewarded (e.g. [39,49]).

Individuals who invested the highest amount in the third-party stage were preferentially chosen as partners, although this preference was more pronounced in the help than in the punishment condition. Third-party investments in punishment and helping were also positively associated with the amount that bystanders entrusted to them, suggesting that these investments increased perceived trustworthiness (as in [15]). Investments in third-party punishment and third-party helping were also both positively associated with actual trustworthiness, which suggests that the perceptions of these individuals' trustworthiness were justified.

Nevertheless, we also uncovered several differences in how third-party signals in help and punishment were interpreted by bystanders. For instance, while lowest investors were almost never chosen by bystanders in the help condition, lowest investors were often chosen as partners in the punishment condition. Moreover, players who invested nothing in third-party help were less trustworthy than players who invested nothing in third-party punishment. These results suggest that doing nothing sent a different signal in the help versus the punishment condition. Individuals who do not help are assumed to be (and actually are) less trustworthy, whereas the same is not true for individuals who do not punish. Thus, while third-party punishers may be less likely than third-party helpers to accrue a positive reputation, these results also imply that a failure to help is more damaging to one's reputation than a failure to punish.

Our results can help distinguish between two reputation-based accounts for the evolution of punishment: indirect reciprocity versus signalling within a system of reputation-based partner choice (for the difference, see [50]). Signalling theory predicts that observers will place more weight on more informative signals, such that the reputational benefits for an action depend on how well that action correlates with whatever trait is being

signalled. In our study, helping behaviour correlated better with trustworthiness than punishment did, and bystanders correspondingly based their partner choice more on helping behaviour than on punishment (i.e. they relied more on the more informative signal). Given this difference in reputational benefits, third-parties invested more in helping than in punishment (i.e. they invested more in the better signal). This is exactly what one predicts from a signalling account of punishment whereby punishment affects reputation-based partner choice. It is less clear how indirect reciprocity would account for the differential rewards for helping versus punishment and the differential honesty of helping versus punishment. As such, our results provide better support for reputation-based partner choice than indirect reciprocity (see also [50]).

Although many of our results align with previous results on helping and punishment, we note that we did not replicate earlier results reported in Barclay & Willer [27] and Sylwester & Roberts [31]. In those studies, the potential for partner choice caused an escalation of investments in costly helpful behaviour, relative to observation alone (see also [23]). In our study, third-party investment did not significantly differ between conditions where behaviour was observable (but participants could not be chosen on the basis of their behaviour) and conditions where players could additionally be chosen on the basis of their investments. We note one discrepancy between our study and the previous ones that might explain the differences in results. In Barclay & Willer [27] and Sylwester & Roberts [31], participants chosen as partners subsequently played a Mutual Aid Game where they initially received a new endowment. Being chosen as partner for the second interaction, therefore, yielded direct access to new resources, as it did in some other studies showing evidence of competitive giving [51,52] but see [53]. Conversely, in our design, chosen participants took part in a Trust Game as trustees and did not necessarily receive any resources from the new partner (because trusters could choose to send no money to the participant in the Trust Game). In this study, therefore, investments in costly signalling in the first interaction were imbued (perhaps

more realistically) with a higher level of uncertainty regarding the benefits of being chosen as a partner. Similarly, any gains from the Trust Game might not have been enough to motivate participants to compete to help or punish, especially if they intended to be trustworthy and return half of what they were entrusted with. We think that further exploring the potential for partner selection to generate competitive altruism in more realistic settings (e.g. where the benefit of being selected as social partner is not a fixed guaranteed reward) is an interesting direction for future research.

We would like to highlight several limitations to this study. This study was conducted using an online platform and a predominantly Western sample—and, as such, the usual caveats on generalizability should be borne in mind. We also note that comprehension was relatively low and that our sample size was substantially reduced by excluding players who failed relevant comprehension checks. We also regret that our analyses were not pre-registered and, while our main hypotheses were generated *a priori*, several analytical decisions were taken after we had already collected the data and should be interpreted with this in mind.

A further limitation is that this study investigated whether costly investments in punishment and helping behaviour would be increased when there was a potential for partner choice in a highly abstract and decontextualized setting—and where the individual was being chosen for a future task involving cooperation. While the decontextualized nature of the economic game we used can reveal the baseline judgements of, and preferences to interact with, third-party punishers and helpers, we stress that these results may not generalize to other abstract scenarios or to all real-world settings. For example, people may believe that punishment is more appropriate than compensating victims in other settings [54]; and punishers may sometimes be approved of and preferentially chosen over helpers for interactions. This may be especially likely when evaluating the qualities needed to be a tough and competent leader and making hypothetical leader choices. Indeed, people evaluate more positively and preferentially choose a punisher over a compensator for a leadership role when this is the case [37]. Similarly, under economic uncertainty or scenarios involving conflict, people may also prefer dominant or authoritarian leaders over prestigious leaders [55,56]. Nevertheless, while we might expect a stronger preference for punitive over helpful partners (or leaders) in some settings, this does not necessarily mean that we would observe competition to be increasingly punitive in such settings. Whether potential leaders escalate their investments in punishment when there is pressure to be seen as tough and competent is an open question for future research.

In conclusion, our results indicate that reputation-based partner choice may support the evolution of third-party punishment by allowing punishers to recoup the costs associated with punishment via increased access to cooperative interactions. These findings therefore have implications for the evolution of such costly strategies. Our results are less clear on the idea that people use investments in punishment or help to compete with others to be chosen as interaction partners and we look forward to further work that will help to address this issue.

Finally, our results support the conceptualization of punishment as a more ambiguous signal of cooperative intent [16] compared to helping behaviour, and may also help to explain why our participants preferred to invest in helpful rather than punitive behaviour—a finding which replicates previous results [17,42]. In keeping with the hypothesis that punishment is an ambiguous signal of cooperative intent, our results also suggest that investments in third-party help more reliably signal trustworthiness than investments in third-party punishment and, accordingly, observers have a stronger preference to interact with more helpful compared to more punitive partners. Importantly, it also seems apparent from this (and previous) work that whereas most cooperative individuals invest to help others, not all are willing to invest in third-party punishment. Understanding the roots of this variance (why do some cooperative individuals invest in punishment, while others do not?) is an important direction for future research.

**Ethics.** This study was approved by the UCL Research Ethics Committee (project 3720/001).

**Data accessibility.** All data, code and materials to reproduce this study can be found at https://osf.io/4zpkb/.

**Authors' contributions.** T.B.: conceptualization, data curation, formal analysis, investigation, methodology, project administration, writing—original draft, writing—review and editing; P.B.: conceptualization, formal analysis, investigation, methodology, writing—original draft, writing—review and editing; N.J.R.: conceptualization, formal analysis, funding acquisition, investigation, methodology, project administration, supervision, visualization, writing—original draft, writing—review and editing. All authors gave final approval for publication and agreed to be held accountable for the work performed therein.

**Competing interests.** We declare we have no competing interests.

**Funding.** P.B. is supported by the Social Sciences and Humanities Research Council of Canada (SSHRC grant no. 430287). N.J.R. is supported by the Royal Society and the Leverhulme Trust.

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
