## [Peer Review File · Proceedings of the Royal Society B: Biological Sciences]

Review History

RSPB-2021-1773.R0 (Original submission)

Review form: Reviewer 1

Recommendation

Major revision is needed (please make suggestions in comments)

Scientific importance: Is the manuscript an original and important contribution to its field?

Good

General interest: Is the paper of sufficient general interest?

Good

Quality of the paper: Is the overall quality of the paper suitable?

Good

Is the length of the paper justified?

Yes

Should the paper be seen by a specialist statistical reviewer?

No

Do you have any concerns about statistical analyses in this paper? If so, please specify them explicitly in your report.

No

It is a condition of publication that authors make their supporting data, code and materials available - either as supplementary material or hosted in an external repository. Please rate, if applicable, the supporting data on the following criteria.

Is it accessible?

Yes

Is it clear?

Yes

Is it adequate?

Yes

Do you have any ethical concerns with this paper?

No

Comments to the Author

In “Third-party punishers do not compete to be chosen as partners” the authors find across one experiment that partner choice competition does not increase investments in third-party punishment over and above the effect of non-anonymity. They also find that third-party punishers are seen as more trustworthy than third-parties who do not respond, but less trustworthy than third-party helpers. They also find that third-party punishment is an honest signal of trustworthiness, but third-party helping is an even more dependable signal of trustworthiness.

This is a well-executed, clearly written paper. I enjoyed reading it. While some of the research findings have been already demonstrated in previous research (e.g., Dhaliwal, Patil, & Cushman, 2021; Raihani & Bshary, 2015), the question regarding the effect of partner choice competition on investments in third-party punishment I believe is novel and interesting. And the unexpected finding regarding those who do not invest in third-party helping and those who do not invest in third-party punishment is very interesting as well. With that said, I do have some points that I think could be addressed to improve the manuscript.

I would appreciate it if the authors could make it clearer how this research adds to our understanding of the evolution of cooperation. What does the null effect of partner choice on third-party punishment imply for the evolution of third-party punishment via indirect reciprocity? And how does the comparison between punishing and helping assist in adjudicating between different ultimate explanations regarding the cooperation?

Could the authors also explain why they chose to compare punishment with helping, since helping does not incentivize cooperation. Why not instead compare punishment with say, rewarding those who choose to cooperate, given that both punishment and rewarding both help promote cooperation? Some clarity on this point would be appreciated.

I think it would be worthwhile for the authors to develop their idea a bit more for why punishment is a weaker signal of trustworthiness compared to helping. Why does such a signal exist? Is it in some way less costly, or more beneficial, for more cooperative types to choose to help rather than punish when in the position of a third-party responder?

The authors note on p. 3, “the context in which punishment occurs is likely to determine the information it conveys and, therefore, its reputation consequences”. I agree with the authors, and thus I’m concerned that the narrow, decontextualized nature of economic games may be

leading to an incomplete portrayal of the phenomenon being examined. Could the authors explain whether they predict that in more “real life” situations, choosing to help, rather than engage in the often more difficult behaviour of punishment could be seen as taking the easy way out and failing to fully confront and deal with the issue. And if the authors agree, what does this imply for the importance and generalizability of their findings?

I understand that the authors made punishment and helping equally costly in the economic game, but that doesn't erase the fact that in everyday life, punishment is often costlier. And even when it is not costlier, it is often seen by victims as the proper response to many types of violations (e.g., Reb, Goldman, Kray, & Cropanzano, 2006), as I expect would also be the case from the perspective of bystanders. For example, would people see a school principal as being trustworthy if he chose to hold off from punishing a bully and instead just provide some compensation to the victim? Would people see a judge as being trustworthy if she chose to not punish a criminal and instead just provide some help to the victim? If someone was attacked in the street, would people see a bystander as being more trustworthy if he chose to not chase down the attacker and instead attend to the victim?

I appreciate that economic games provide a means to observe real incentivized behaviour in online experiments, but the lack of context I believe leads to some attributional ambiguity when it comes to judging a punisher. The authors state that punishment is a more “ambiguous signal of cooperative intent”, but might this ambiguity be partly due to much of the research comparing punishing to helping being done with decontextualized economic games?

I'm curious to know whether the authors believe their null effect for partner choice might also be a product of the decontextualized nature of an economic game. Do the authors think that in a “real life” situation people would still not upregulate their willingness to punish when they have the chance to be chosen as a cooperation partner. In other words, do the authors have a theoretical rationale for why partner choice does not have an effect on third-party punishment, which leads them to believe that this null finding will be observed in other contexts? If not, perhaps the claim that “third-party punishers do not compete to be chosen as partners” is a bit too decisive in tone. Could the authors explain why they choose to frame the paper in such a conclusive way?

The authors twice refer to the “strategy method” without explaining what it entails. I think readers who aren't steeped in the experimental economics literature might benefit from a brief explanation of what this means.

References:

Dhaliwal, N. A., Patil, I., & Cushman, F. (2021). Reputational and cooperative benefits of third-party compensation. *Organizational Behavior and Human Decision Processes*, 164, 27-51.

Raihani, N. J., & Bshary, R. (2015). Third-party punishers are rewarded, but third-party helpers even more so. *Evolution*, 69(4), 993-1003.

Reb, J., Goldman, B. M., Kray, L. J., & Cropanzano, R. (2006). Different wrongs, different remedies? Reactions to organizational remedies after procedural and interactional injustice. *Personnel Psychology*, 59(1), 31-64.

Review form: Reviewer 2

Recommendation

Major revision is needed (please make suggestions in comments)

Scientific importance: Is the manuscript an original and important contribution to its field?
Acceptable

General interest: Is the paper of sufficient general interest?
Good

Quality of the paper: Is the overall quality of the paper suitable?
Good

Is the length of the paper justified?
Yes

Should the paper be seen by a specialist statistical reviewer?
No

Do you have any concerns about statistical analyses in this paper? If so, please specify them explicitly in your report.
Yes

It is a condition of publication that authors make their supporting data, code and materials available - either as supplementary material or hosted in an external repository. Please rate, if applicable, the supporting data on the following criteria.

Is it accessible?
Yes

Is it clear?
Yes

Is it adequate?
No

Do you have any ethical concerns with this paper?
No

Comments to the Author

The current study examined the reputational benefits of third-party punishment (vs. third-party helping). The study was well-designed with clear research questions. I especially like that the fee-to-fine ratio was 1:1 in punishment, which allowed the authors to test the "punishment as a cooperative intent signal" more strictly. I have several concerns that need to be addressed.

- It was nice that the current findings provide converging evidence that helping is preferred over punishment in general (e.g., Jordan et al., 2016, Dhaliwal et al., 2021). However, it was unclear what the unique theoretical contribution of the current study is. Maybe authors can write a little bit more about how the current study complements the existing literature in the Introduction more. Also, I think the authors should cite the paper below which is very relevant to the current study:

Dhaliwal, N. A. *, Patil, I. *, & Cushman, F. A. (2021). Reputational and cooperative benefits of third-party compensation. *Organizational Behavior and Human Decision Processes*, 164, 27-51.

- I think the authors should explain more clearly what they manipulated in "Random vs. Partner Choice" and "Knowledge vs. No knowledge". It was unclear to what extent the third party was aware of the experimental setting. In the Knowledge condition, were third parties aware of the fact that bystanders will be informed of how the third parties behaved in the Help/Punishment stage? Also, were third parties aware of the fact that the allocation will be random or partner choice? I thought that the Knowledge condition involved whether the bystander was informed of

the third party's behavior (line #177), but was confused when it sounded like the condition was actually about the third party punisher's knowledge. Please clarify your experimental manipulations. Also, it would be easier for readers to follow if authors explain expected results between conditions more clearly at the end of the Introduction section and explain the rationale why you varied Knowledge and Allocation so that readers can understand better what you tried to disentangle by manipulating these variables.

- I have several clarifying questions about the data analyses. First, why did the authors convert a continuous dependent variable into a categorical one when testing H1, 2, and 4? I think the continuous DV is more fine-grained, and I couldn't find a clear reason for the decision. Could authors include the analyses with continuous DV in Supplementary Material? And/or explain more clearly why you had to turn continuous measure into a categorical one with arbitrary criteria?

- Second, I appreciate that the authors reported various analyses with the full and reduced sample. However, as a reader, it is very confusing to learn different results from Bayesian vs. frequentist and full vs. reduced sample. If possible, could the authors focus on one method (Bayesian vs. frequentist) and one sample (full vs. reduced) with clear rationale in the main text and move the rest to SOM? Relatedly, it is very confusing in the Results section (especially those from H1) because the results are slightly different across analyses. Could the authors end the results section (e.g., at the end of H1 results) with a summary sentence where you highlight the common patterns across different analyses, so that readers can digest the results more easily?

- Third, I don't understand the sentence here "In the punishment condition, 44 participants (22 pairs) pairs invested the same amount in punishment and could not be used in this analysis." on page 13. Why can't they be analyzed? Could you elaborate on it? Further, I was confused why you used binomial tests here. Wouldn't it be more straightforward to run some sort of correlational test or GLM (e.g., probability of being chosen as a partner ~ invest.amount*condition) in which you don't have to throw away a part of your data.

- Lastly, could you explain your prior distributions of Bayesian analyses? From my understanding, the results can be different depending on which prior distribution you set for your model. Or are your models robust against different prior distributions?

- The authors wrote that all data and code are available at <https://osf.io/4zpkb/> but I couldn't find them.

Decision letter (RSPB-2021-1773.R0)

28-Sep-2021

Dear Professor Raihani:

Your manuscript has now been peer reviewed and the reviews have been assessed by an Associate Editor. The reviewers' comments (not including confidential comments to the Editor) and the comments from the Associate Editor are included at the end of this email for your reference. As you will see, the reviewers and the Editors have raised some concerns with your manuscript and we would like to invite you to revise your manuscript to address them.

We do not allow multiple rounds of revision so we urge you to make every effort to fully address all of the comments at this stage. If deemed necessary by the Associate Editor, your manuscript will be sent back to one or more of the original reviewers for assessment. If the original reviewers

are not available we may invite new reviewers. Please note that we cannot guarantee eventual acceptance of your manuscript at this stage.

Research ethics:

Use of animals and field studies:

It is a condition of publication that you make available the data and research materials supporting the results in the article. Please see our Data Sharing Policies (<https://royalsociety.org/journals/authors/author-guidelines/#data>). Datasets should be deposited in an appropriate publicly available repository and details of the associated accession number, link or DOI to the datasets must be included in the Data Accessibility section of the article (<https://royalsociety.org/journals/ethics-policies/data-sharing-mining/>). Reference(s) to datasets should also be included in the reference list of the article with DOIs (where available).

Please submit a copy of your revised paper within three weeks. If we do not hear from you within this time your manuscript will be rejected. If you are unable to meet this deadline please let us know as soon as possible, as we may be able to grant a short extension.

Best wishes,

Professor Gary Carvalho

Associate Editor

Board Member: 1

Comments to Author:

The reviewers and I agree that this is an interesting, well-designed behavioral study. In addition to their thoughtful comments about the study's framing, methods, and analyses, I would like to emphasize that I agree with a key point raised by both reviewers: the paper currently lacks a clear discussion and framing in terms of larger theoretical implications. In particular, the paper currently reads as more appropriate for a psychology or behavioral economics journal, but the stakes of these questions need to be made apparent to a general biological readership. For example, R1 notes that it would be important to more clearly discuss how this work informs our understanding of the evolution of cooperation. R2 further emphasizes that it would be important to be clearer about what this adds to prior work on this topic. Related to these general concerns, I think it would improve the paper to make a clearer link between this (particular, narrow) experimental game context, and real-life situations where these kinds of decisions about whether to help, punish, and engage in partner choice are relevant.

Reviewer(s)' Comments to Author:

Referee: 1

Comments to the Author(s)

In "Third-party punishers do not compete to be chosen as partners" the authors find across one experiment that partner choice competition does not increase investments in third-party punishment over and above the effect of non-anonymity. They also find that third-party punishers are seen as more trustworthy than third-parties who do not respond, but less trustworthy than third-party helpers. They also find that third-party punishment is an honest signal of trustworthiness, but third-party helping is an even more dependable signal of trustworthiness.

This is a well-executed, clearly written paper. I enjoyed reading it. While some of the research findings have been already demonstrated in previous research (e.g., Dhaliwal, Patil, & Cushman, 2021; Raihani & Bshary, 2015), the question regarding the effect of partner choice competition on investments in third-party punishment I believe is novel and interesting. And the unexpected finding regarding those who do not invest in third-party helping and those who do not invest in

third-party punishment is very interesting as well. With that said, I do have some points that I think could be addressed to improve the manuscript.

I would appreciate it if the authors could make it clearer how this research adds to our understanding of the evolution of cooperation. What does the null effect of partner choice on third-party punishment imply for the evolution of third-party punishment via indirect reciprocity? And how does the comparison between punishing and helping assist in adjudicating between different ultimate explanations regarding the cooperation?

Could the authors also explain why they chose to compare punishment with helping, since helping does not incentivize cooperation. Why not instead compare punishment with say, rewarding those who choose to cooperate, given that both punishment and rewarding both help promote cooperation? Some clarity on this point would be appreciated.

I think it would be worthwhile for the authors to develop their idea a bit more for why punishment is a weaker signal of trustworthiness compared to helping. Why does such a signal exist? Is it in some way less costly, or more beneficial, for more cooperative types to choose to help rather than punish when in the position of a third-party responder?

The authors note on p. 3, "the context in which punishment occurs is likely to determine the information it conveys and, therefore, its reputation consequences". I agree with the authors, and thus I'm concerned that the narrow, decontextualized nature of economic games may be leading to an incomplete portrayal of the phenomenon being examined. Could the authors explain whether they predict that in more "real life" situations, choosing to help, rather than engage in the often more difficult behaviour of punishment could be seen as taking the easy way out and failing to fully confront and deal with the issue. And if the authors agree, what does this imply for the importance and generalizability of their findings?

I understand that the authors made punishment and helping equally costly in the economic game, but that doesn't erase the fact that in everyday life, punishment is often costlier. And even when it is not costlier, it is often seen by victims as the proper response to many types of violations (e.g., Reb, Goldman, Kray, & Cropanzano, 2006), as I expect would also be the case from the perspective of bystanders. For example, would people see a school principal as being trustworthy if he chose to hold off from punishing a bully and instead just provide some compensation to the victim? Would people see a judge as being trustworthy if she chose to not punish a criminal and instead just provide some help to the victim? If someone was attacked in the street, would people see a bystander as being more trustworthy if he chose to not chase down the attacker and instead attend to the victim?

I appreciate that economic games provide a means to observe real incentivized behaviour in online experiments, but the lack of context I believe leads to some attributional ambiguity when it comes judging a punisher. The authors state that punishment is a more "ambiguous signal of cooperative intent", but might this ambiguity be partly due to much of the research comparing punishing to helping being done with decontextualized economic games?

I'm curious to know whether the authors believe their null effect for partner choice might also be a product of the decontextualized nature of an economic game. Do the authors think that in a "real life" situation people would still not upregulate their willingness to punish when they have the chance to be chosen as a cooperation partner. In other words, do the authors have a theoretical rationale for why partner choice does not have an effect on third-party punishment, which leads them to believe that this null finding will be observed in other contexts? If not, perhaps the claim that "third-party punishers do not compete to be chosen as partners" is a bit too decisive in tone. Could the authors explain why they chose to frame the paper in such a conclusive way?

The authors twice refer to the “strategy method” without explaining what it entails. I think readers who aren’t steeped in the experimental economics literature might benefit from a brief explanation of what this means.

References:

Dhaliwal, N. A., Patil, I., & Cushman, F. (2021). Reputational and cooperative benefits of third-party compensation. *Organizational Behavior and Human Decision Processes*, 164, 27-51.

Raihani, N. J., & Bshary, R. (2015). Third-party punishers are rewarded, but third-party helpers even more so. *Evolution*, 69(4), 993-1003.

Reb, J., Goldman, B. M., Kray, L. J., & Cropanzano, R. (2006). Different wrongs, different remedies? Reactions to organizational remedies after procedural and interactional injustice. *Personnel Psychology*, 59(1), 31-64.

Referee: 2

Comments to the Author(s)

The current study examined the reputational benefits of third-party punishment (vs. third-party helping). The study was well-designed with clear research questions. I especially like that the fee-to-fine ratio was 1:1 in punishment, which allowed the authors to test the "punishment as a cooperative intent signal" more strictly. I have several concerns that need to be addressed.

- It was nice that the current findings provide converging evidence that helping is preferred over punishment in general (e.g., Jordan et al., 2016, Dhaliwal et al., 2021). However, it was unclear what the unique theoretical contribution of the current study is. Maybe authors can write a little bit more about how the current study complements the existing literature in the Introduction more. Also, I think the authors should cite the paper below which is very relevant to the current study:

Dhaliwal, N. A.*, Patil, I.*, & Cushman, F. A. (2021). Reputational and cooperative benefits of third-party compensation. *Organizational Behavior and Human Decision Processes*, 164, 27-51.

- I think the authors should explain more clearly what they manipulated in "Random vs. Partner Choice" and "Knowledge vs. No knowledge". It was unclear to what extent the third party was aware of the experimental setting. In the Knowledge condition, were third parties aware of the fact that bystanders will be informed of how the third parties behaved in the Help/Punishment stage? Also, were third parties aware of the fact that the allocation will be random or partner choice? I thought that the Knowledge condition involved whether the bystander was informed of the third party's behavior (line #177), but was confused when it sounded like the condition was actually about the third party punisher's knowledge. Please clarify your experimental manipulations. Also, it would be easier for readers to follow if authors explain expected results between conditions more clearly at the end of the Introduction section and explain the rationale why you varied Knowledge and Allocation so that readers can understand better what you tried to disentangle by manipulating these variables.

- I have several clarifying questions about the data analyses. First, why did the authors convert a continuous dependent variable into a categorical one when testing H1, 2, and 4? I think the continuous DV is more fine-grained, and I couldn't find a clear reason for the decision. Could authors include the analyses with continuous DV in Supplementary Material? And/or explain more clearly why you had to turn continuous measure into a categorical one with arbitrary criteria?

- Second, I appreciate that the authors reported various analyses with the full and reduced sample. However, as a reader, it is very confusing to learn different results from Bayesian vs. frequentist and full vs. reduced sample. If possible, could the authors focus on one method (Bayesian vs. frequentist) and one sample (full vs. reduced) with clear rationale in the main text and move the rest to SOM? Relatedly, it is very confusing in the Results section (especially those

from H1) because the results are slightly different across analyses. Could the authors end the results section (e.g., at the end of H1 results) with a summary sentence where you highlight the common patterns across different analyses, so that readers can digest the results more easily?

- Third, I don't understand the sentence here "In the punishment condition, 44 participants (22 pairs) pairs invested the same amount in punishment and could not be used in this analysis." on page 13. Why can't they be analyzed? Could you elaborate on it? Further, I was confused why you used binomial tests here. Wouldn't it be more straightforward to run some sort of correlational test or GLM (e.g., probability of being chosen as a partner \sim invest.amount*condition) in which you don't have to throw away a part of your data.

- Lastly, could you explain your prior distributions of Bayesian analyses? From my understanding, the results can be different depending on which prior distribution you set for your model. Or are your models robust against different prior distributions?

- The authors wrote that all data and code are available at <https://osf.io/4zpkb/> but I couldn't find them.

Author's Response to Decision Letter for (RSPB-2021-1773.R0)

See Appendix A.

RSPB-2021-1773.R1 (Revision)

Review form: Reviewer 1

Recommendation

Accept with minor revision (please list in comments)

Scientific importance: Is the manuscript an original and important contribution to its field?

Good

General interest: Is the paper of sufficient general interest?

Good

Quality of the paper: Is the overall quality of the paper suitable?

Good

Is the length of the paper justified?

Yes

Should the paper be seen by a specialist statistical reviewer?

No

Do you have any concerns about statistical analyses in this paper? If so, please specify them explicitly in your report.

No

It is a condition of publication that authors make their supporting data, code and materials available - either as supplementary material or hosted in an external repository. Please rate, if applicable, the supporting data on the following criteria.

Is it accessible?

Yes

Is it clear?

Yes

Is it adequate?

Yes

Do you have any ethical concerns with this paper?

No

Comments to the Author

The manuscript is much improved. That said, I think it would be helpful and interesting to the readers of this paper if the authors could tie everything together by explaining in a bit more detail what this collection of results might imply for the evolution of third-party punishment via indirect reciprocity.

Review form: Reviewer 2

Recommendation

Accept with minor revision (please list in comments)

Scientific importance: Is the manuscript an original and important contribution to its field?

Good

General interest: Is the paper of sufficient general interest?

Excellent

Quality of the paper: Is the overall quality of the paper suitable?

Good

Is the length of the paper justified?

Yes

Should the paper be seen by a specialist statistical reviewer?

No

Do you have any concerns about statistical analyses in this paper? If so, please specify them explicitly in your report.

No

It is a condition of publication that authors make their supporting data, code and materials available - either as supplementary material or hosted in an external repository. Please rate, if applicable, the supporting data on the following criteria.

Is it accessible?

Yes

Is it clear?

Yes

Is it adequate?

Yes

Do you have any ethical concerns with this paper?

No

Comments to the Author

I thank the authors for their reply to my comments and concerns. I believe their manuscript is much improved and clear now. I just have a few suggestions for final edits.

I think the Discussion in the current manuscript is very descriptive (like a summary of results rather than a discussion of theoretical implications). It would be helpful for readers to highlight some theoretical implications of this work. One potential way to do this is by connecting the current findings with some hypotheses mentioned in the Intro. Personally, I'm interested in hearing about how the current findings inform the hypotheses in the Introduction e.g., "An alternative route by which punitive strategies could yield individual-level benefits to punishers is via reputation consequences that increase the punisher's likelihood to have profitable social interactions in the future: (i) by signalling formidability or (ii) by signalling cooperative intent." (line 45 on p. 2).

Just to clarify, in the Helping condition, third parties were able to choose between helping a receiver and doing nothing (i.e., no investment). Similarly, in the Punishment condition, they chose between punishing a dictator and doing nothing (i.e., no investment). Is this correct? If so, some descriptions in the Method may be confusing to readers as it sounded like participants chose between helping and punishment, which was actually a between-subject variable. For example, from line 193 on p. 6, the authors wrote "Following the dictator decision, the third-party chose how much of their endowment, if any, to invest to punish an unfair Dictator or, according to the experimental condition, to help a receiver who was given an unfair share. Third-parties were endowed with \$0.50 and could invest any amount (in \$0.01 increments) between \$0.00 and \$0.45 to either help or punish the target individual, with a fee to fine/fee to help ratio of 1:1". It would be helpful for readers to be clear about the fact that there was a "no investment" option in both conditions, so the readers don't misunderstand helping vs. punishment as a within-subject variable.

The authors called the current finding a "failed replication" (line 439 on p.15), which has a negative connotation. I would be more cautious about viewing this study as a failed replication because as the authors noted, there were differences in study designs between the two studies. So replacing "this failed replication" with "differences in results" or something similar would be more appropriate in this context.

Decision letter (RSPB-2021-1773.R1)

17-Nov-2021

Dear Professor Raihani:

Your manuscript has now been peer reviewed and the reviews have been assessed by an Associate Editor. The reviewers' comments (not including confidential comments to the Editor) and the comments from the Associate Editor are included at the end of this email for your reference. As you will see, the reviewers and the Editors have raised some concerns with your manuscript and we would like to invite you to revise your manuscript to address them.

Research ethics:

Use of animals and field studies:

It is a condition of publication that you make available the data and research materials supporting the results in the article (<https://royalsociety.org/journals/authors/author-guidelines/#data>). Datasets should be deposited in an appropriate publicly available repository and details of the associated accession number, link or DOI to the datasets must be included in the Data Accessibility section of the article (<https://royalsociety.org/journals/ethics-policies/data-sharing-mining/>). Reference(s) to datasets should also be included in the reference list of the article with DOIs (where available).

Please submit a copy of your revised paper within three weeks. If we do not hear from you within this time your manuscript will be rejected. If you are unable to meet this deadline please let us know as soon as possible, as we may be able to grant a short extension.

Best wishes,
Professor Gary Carvalho
Editor, Proceedings B
mailto:proceedingsb@royalsociety.org

Associate Editor
Board Member: 1
Comments to Author:

The reviewers and I both agree that this is a responsive review that addresses most of the points raised in the prior review. R1 one has a clarification point on methods that should be addressed. My primary feedback, in line with points raised by both R1 and R2, concerns the intellectual framing of the paper and discussion. For example, R2 mentions the importance of articulating the implications of the results for the evolution of third-party punishment via indirect reciprocity, and R1 notes that the discussion is currently fairly descriptive rather than theory-driven. I agree with these points. For example, not explicitly addressing theoretical ideas concerning whether punishment versus help are better signals of cooperative intent in the introduction (the response letter mentions that this is because this is addressed in other papers) leaves the paper reading as more narrow and less relevant to a broader biological audience. It also means that some aspects of the paper, especially those looking at whether costly helping is treated as a signal of cooperative intent (introduced on pages 4-5), are motivated more terms of methodological rather than theoretical questions. Given that the direct contrast between the efficacy of punishment and helping as a signal is a key part of the study design and results, the theoretical reason this is interesting needs to be made clear in both the introduction and results.

Reviewer(s)' Comments to Author:
Referee: 2

Comments to the Author(s)

I thank the authors for their reply to my comments and concerns. I believe their manuscript is much improved and clear now. I just have a few suggestions for final edits.

I think the Discussion in the current manuscript is very descriptive (like a summary of results rather than a discussion of theoretical implications). It would be helpful for readers to highlight some theoretical implications of this work. One potential way to do this is by connecting the

current findings with some hypotheses mentioned in the Intro. Personally, I'm interested in hearing about how the current findings inform the hypotheses in the Introduction e.g., "An alternative route by which punitive strategies could yield individual-level benefits to punishers is via reputation consequences that increase the punisher's likelihood to have profitable social interactions in the future: (i) by signalling formidability or (ii) by signalling cooperative intent." (line 45 on p. 2).

Just to clarify, in the Helping condition, third parties were able to choose between helping a receiver and doing nothing (i.e., no investment). Similarly, in the Punishment condition, they chose between punishing a dictator and doing nothing (i.e., no investment). Is this correct? If so, some descriptions in the Method may be confusing to readers as it sounded like participants chose between helping and punishment, which was actually a between-subject variable. For example, from line 193 on p. 6, the authors wrote "Following the dictator decision, the third-party chose how much of their endowment, if any, to invest to punish an unfair Dictator or, according to the experimental condition, to help a receiver who was given an unfair share. Third-parties were endowed with \$0.50 and could invest any amount (in \$0.01 increments) between \$0.00 and \$0.45 to either help or punish the target individual, with a fee to fine/fee to help ratio of 1:1". It would be helpful for readers to be clear about the fact that there was a "no investment" option in both conditions, so the readers don't misunderstand helping vs. punishment as a within-subject variable.

The authors called the current finding a "failed replication" (line 439 on p.15), which has a negative connotation. I would be more cautious about viewing this study as a failed replication because as the authors noted, there were differences in study designs between the two studies. So replacing "this failed replication" with "differences in results" or something similar would be more appropriate in this context.

Referee: 1

Comments to the Author(s)

The manuscript is much improved. That said, I think it would be helpful and interesting to the readers of this paper if the authors could tie everything together by explaining in a bit more detail what this collection of results might imply for the evolution of third-party punishment via indirect reciprocity.

Author's Response to Decision Letter for (RSPB-2021-1773.R1)

See Appendix B.

Decision letter (RSPB-2021-1773.R2)

07-Dec-2021

Dear Professor Raihani

I am pleased to inform you that your manuscript entitled "Third-party punishers do not compete to be chosen as partners in an experimental game" has been accepted for publication in Proceedings B.

Data Accessibility section

Open Access

Paper charges

Sincerely,

Professor Gary Carvalho

Associate Editor:

Comments to Author:

The revision has addressed the outstanding comments from the prior round of review.

Appendix A

Associate Editor

Board Member: 1

Comments to Author:

The reviewers and I agree that this is an interesting, well-designed behavioral study. In addition to their thoughtful comments about the study's framing, methods, and analyses, I would like to emphasize that I agree with a key point raised by both reviewers: the paper currently **lacks a clear discussion and framing in terms of larger theoretical implications**. In particular, the paper currently reads as more appropriate for a psychology or behavioral economics journal, but the stakes of these questions need to be made apparent to a general biological readership. For example, R1 notes that it would be important to more clearly discuss how this work informs our understanding of the evolution of cooperation. R2 further emphasizes that it would be important to be clearer about what this adds to prior work on this topic. Related to these general concerns, I think it would improve the paper to make a clearer link between this (particular, narrow) experimental game context, and real-life situations where these kinds of decisions about whether to help, punish, and engage in partner choice are relevant.

Thanks for the generally positive comments on the manuscript and for the opportunity to revise and resubmit.

We have taken the helpful comments from the reviewers on board and respond to them point by point (in red) below.

Any changes to the manuscript text are shown in blue.

Reviewer(s)' Comments to Author:

Referee: 1

Comments to the Author(s)

In “Third-party punishers do not compete to be chosen as partners” the authors find across one experiment that partner choice competition does not increase investments in third-party punishment over and above the effect of non-anonymity. They also find that third-party punishers are seen as more trustworthy than third-parties who do not respond, but less trustworthy than third-party helpers. They also find that third-party punishment is an honest signal of trustworthiness, but third-party helping is an even more dependable signal of trustworthiness.

This is a well-executed, clearly written paper. I enjoyed reading it. While some of the research findings have been already demonstrated in previous research (e.g., Dhaliwal, Patil, & Cushman, 2021; Raihani & Bshary, 2015), the question regarding the effect of partner choice competition on investments in third-party punishment I believe is novel and interesting. And the unexpected finding regarding those who do not invest in third-party helping and those who do not invest in third-party punishment is very interesting as well. With that said, I do have some points that I think could be addressed to improve the manuscript.

Thank you for the positive assessment and for alerting us to the important Dhaliwal et al. paper, which we ought to have cited (and now do).

I would appreciate it if the authors could make it clearer how this research adds to our understanding of the evolution of cooperation. What does the null effect of partner choice on third-party punishment imply for the evolution of third-party punishment via indirect reciprocity? And how does the comparison between punishing and helping assist in adjudicating between different ultimate explanations regarding the cooperation?

Could the authors also explain why they chose to compare punishment with helping, since helping does not incentivize cooperation. Why not instead compare punishment with say, rewarding those who choose to cooperate, given that both punishment and rewarding both help promote cooperation? Some clarity on this point would be appreciated.

While we acknowledge that lots of work has been done to understand the potential signalling benefits of third-party punishment, our specific question was to test a prediction from costly signalling theory which is that investments into costly signals (of punishment) ought to increase when there is competition to be chosen by a partner.

We chose to compare helping and punishment because we wanted to run a conceptual replication of Barclay & Willer 2007, which reported evidence for competitive helping in the context of an economic game. In their study, Barclay & Willer also used helping as the putative costly signal which could be used to attract partners. Specifically, our question is whether investments to compensate a victim or to punish a cheat escalate when there is the possibility to be chosen by another partner. We have now made this clearer in a substantially revised version of the introduction, where we state:

A simple hypothesis that does not take ambiguity and context-specificity into account might posit that if punishers stand to gain reputation benefits from punishing, then they should invest more in punishment when these decisions will be made known to other individuals. Similarly, we might expect punishers to further escalate their investments when there is competition to be selected as a social partner, as it has been previously shown for helping behaviour (Willer & Barclay, 2007). This hypothesis might, however, be unsupported if punishment is an ambiguous signal of cooperative intent and if increasing investments in punishment cast further doubt on the punisher's underlying intentions and commitment to cooperation. In this study, therefore, we asked whether punishers do escalate their investments in punishment when there is the potential to be chosen by an observer – and whether punishers are preferred as interaction partners and entrusted with more money. We also examine whether investments in punishment act as honest signals of cooperative intent, by measuring whether punitive investments are related to trustworthy behaviour in a game where individuals are financially incentivised to exploit a partner.

We aimed to test the hypothesis that punishment might be used as a signal of cooperative intent under the most conducive conditions, by using an experimental setting that reduced the potential ambiguities discussed above. As such, we explored the potential signalling value of punishment in a third-party punishment paradigm, where there is a stronger theoretical argument for punishment to operate as a signal of cooperative intent (Heffner and FeldmanHall, 2019; Raihani and Bshary, 2015a). Furthermore, we used a fee-to-fine ratio of 1:1, meaning that punishers could not use punishment as a means to elevate their own payoffs relative to those of the target. This should reduce the possibility to infer that punishment is driven by spiteful preferences rather than an impartial concern for fairness (Raihani and Bshary, 2019, 2015a).

In addition to exploring whether individuals signal cooperative intent using punishment, we also aimed to replicate previous findings that costly investments in helpful behaviours also act as signals of cooperative intent and these investments escalate when there is the possibility to be chosen by a partner for a subsequent social interaction (Barclay and Barker, 2020; Barclay and Willer, 2007; Sylwester and Roberts, 2010).

The prospect of positive reputation gains may increase investments in punishment (or helping) either because the third-party knows that their past behaviour will be revealed to others (who may treat them differently on the basis of previous actions) or because the third-party anticipates to gain access to social partners on the basis of their investment. To distinguish between these two possibilities, we replicate Barclay & Willer's (2007) design by including a 'Knowledge Only' condition, in addition to a 'Knowledge + Partner Choice' condition. This allows us to test whether any increased investment relative to baseline stems from the fact that one's behaviour will be advertised to a potential social partner and whether investments escalate even further when individuals can compete to be chosen as a partner for future social interactions. If third-parties' investments are used to compete to access new social partners, then we expect them to be higher in the Knowledge + Partner Choice condition than they are in the Knowledge Only condition.

We would like to add that we think it is a rather open question as to whether punishment really promotes cooperation, although a full discussion of this lies

beyond the scope of the current paper and has been discussed at length in Raihani & Bshary 2019.

I think it would be worthwhile for the authors to develop their idea a bit more for why punishment is a weaker signal of trustworthiness compared to helping. Why does such a signal exist? Is it in some way less costly, or more beneficial, for more cooperative types to choose to help rather than punish when in the position of a third-party responder?

Thanks for this comment. We have included some additional text to support the intuition that punishment is a weaker signal of cooperative intent than helping behaviour in the revised introduction text (supplied above). One of us had previously outlined this in a review paper, which was cited (Raihani & Bshary 2015, TREE) but we now also cite Dhaliwal et al. 2021 which provides empirical support for the idea.

The authors note on p. 3, “the context in which punishment occurs is likely to determine the information it conveys and, therefore, its reputation consequences”. I agree with the authors, and thus I’m concerned that the narrow, decontextualized nature of economic games may be leading to an incomplete portrayal of the phenomenon being examined. Could the authors explain whether they predict that in more “real life” situations, choosing to help, rather than engage in the often more difficult behaviour of punishment could be seen as taking the easy way out and failing to fully confront and deal with the issue. And if the authors agree, what does this imply for the importance and generalizability of their findings?

Please see the response to the next comment where we have included some additional discussion regarding the decontextualised nature of the game and whether we might expect different results in different contexts.

I understand that the authors made punishment and helping equally costly in the economic game, but that doesn’t erase the fact that in everyday life, punishment is often costlier. And even when it is not costlier, it is often seen by victims as the proper response to many types of violations (e.g., Reb, Goldman, Kray, & Cropanzano, 2006), as I expect would also be the case from the perspective of bystanders. For example, would people see a school principal as being trustworthy if he chose to hold off from punishing a bully and instead just provide some compensation to the victim? Would people see a judge as being trustworthy if she chose to not punish a criminal and instead just provide some help to the victim? If someone was attacked in the street, would people see a bystander as being more trustworthy if he chose to not chase down the attacker and instead attend to the victim?

Thanks for this thoughtful comment. We believe that it is an open empirical question as to whether punishing a cheat or compensating a victim would more often be costlier in real-world settings. For our purposes, it is essential that the costs of each potential investment are held constant, as the information contained in the cost of the signal would confound the information contained in the type of signal if we allowed costs to vary. We have included a paragraph in the discussion about the potential contexts in real life where people might attribute greater reputation benefits to punishers and when punishers might invest more in such actions when they stand to be chosen as interaction partners. We have included the following text in the introduction and discussion, respectively.

Introduction – additional text

The information punishment conveys – and its associated reputation consequences – are likely to be highly context-specific (Raihani and Bshary, 2015a). For example, when individuals are given an option to compensate a victim or to punish a cheat, then punishment is less likely to signal cooperative intent (Dhaliwal et al., 2021; Heffner and FeldmanHall, 2019; Jordan et al., 2016; Raihani and Bshary, 2015b). Contexts where the punisher is harmed directly by the wrongdoer ('second-party punishment') are more likely to be motivated by vengeful sentiments and less likely to be interpreted as signals of the punisher's cooperativeness (Heffner and FeldmanHall, 2019; Raihani and Bshary, 2015a). Conversely, when the punisher is an uninvolved bystander to the initial 'crime' ('third-party punishment'), then punitive acts are more likely to convey cooperative intent. Punishment that is cheaper to administer than the damage it inflicts on a target could also be perceived as a competitive act (Raihani and Bshary, 2019, 2015a), rather than signalling an intent to cooperate. The potential ambiguity and context-dependency of punishment means that, especially in decontextualised settings typical of economic games, the reputation consequences of punishing others might not always be positive (Dhaliwal et al., 2021; Heffner and FeldmanHall, 2019; Horita, 2010; Ozono and Watabe, 2012; Przepiorka and Liebe, 2016; Raihani and Bshary, 2015a). In some cases, therefore, individuals might choose to hide investments in punishing others (Rockenbach and Milinski, 2011) or preferentially invest in helping rather than punishing others when these decisions will be revealed to others (Li et al., 2021). Conversely, because helpful acts entail paying an individual cost to generate benefits to others, the motives underpinning these acts are less likely to be affected by context and may be more likely to be perceived as stemming from genuinely helpful motives (but see Raihani & Power 2021).

Discussion – additional text

A further limitation is that this study investigated whether costly investments in punishment and helping behaviour would be increased when there was a potential for partner choice in a highly abstract and decontextualised setting – and where the individual was being chosen for a future task involving cooperation. While the decontextualised nature of the economic game we used can reveal the baseline judgements of, and preferences to interact with, third-party punishers and helpers, we stress that these results may not generalise to other abstract scenarios

or to all real-world settings. For example, people may believe that punishment is more appropriate than compensating victims in other settings (Reb et al., 2006); and punishers may sometimes be approved of and preferentially chosen over helpers for interactions. This may be especially likely when evaluating the qualities needed to be a tough and competent leader and making hypothetical leader choices. Indeed, people evaluate more positively and preferentially choose a punisher over a compensator for a leadership role when this is the case (Dhaliwal et al., 2021). Similarly, under economic uncertainty or scenarios involving conflict, people may also prefer dominant or authoritarian leaders over prestigious leaders (Kakkar and Sivanathan, 2017; Petersen and Laustsen, 2020). Nevertheless, while we might expect a stronger preference for punitive over helpful partners (or leaders) in some settings, this does not necessarily mean that we would observe competition to be increasingly punitive in such settings. Whether potential leaders escalate their investments in punishment when there is pressure to be seen as tough and competent remains an interesting open question for future research. More broadly, assessing the reputation consequences of punitive acts in real-world, context-rich, scenarios is an important direction for future research.

I appreciate that economic games provide a means to observe real incentivized behaviour in online experiments, but the lack of context I believe leads to some attributional ambiguity when it comes judging a punisher. The authors state that punishment is a more “ambiguous signal of cooperative intent”, but might this ambiguity be partly due to much of the research comparing punishing to helping being done with decontextualized economic games?

Thanks for this interesting question. We agree that these kinds of abstract economic games have limitations (see our point above). It may be the case that punishers could reduce the ambiguity associated with the intentions underlying their decision by (for example) explaining their intentions and goals to observers. Nevertheless, we still think it is the case that helping decisions (whether they are online or enacted in real-world settings) are fundamentally less ambiguous as signals of underlying cooperative intent. This point has been made in Raihani & Bshary (2015) but more clearly tested in Dhaliwal et al. 2021, which we now cite in support of this idea.

I'm curious to know whether the authors believe their null effect for partner choice might also be a product of the decontextualized nature of an economic game. Do the authors think that in a “real life” situation people would still not upregulate their willingness to punish when they have the chance to be chosen as a cooperation partner. In other words, do the authors have a theoretical rationale for why partner choice does not have an effect on third-party punishment, which leads them to believe that this null finding will be observed in other contexts? If not, perhaps the claim that “third-party punishers do not compete to be chosen as partners” is a bit too decisive in tone. Could the authors explain why they choose to frame the paper in such a conclusive way?

We believe we have partially addressed this comment in the section of the discussion mentioned above. Specifically, while it is feasible that punishers will be preferred over helpers as partners (or leaders) in some settings, it is another open question as to whether punishers will compete (by escalating investments in punishment) when there is the possibility to be chosen.

We have changed the title of the paper to reflect the fact that our work only really speaks to the decontextualised nature of an economic game:

Third-party punishers do not compete to be chosen as partners in an experimental game

The authors twice refer to the “strategy method” without explaining what it entails. I think readers who aren’t steeped in the experimental economics literature might benefit from a brief explanation of what this means.

We have included a definition and justification of the strategy method as follows:

The strategy method involves players making conditional decisions for all potential scenarios in an economic game and is widely used in behavioural economics, where it is thought to produce reliable results. (Fischbacher et al., 2012) .

References:

- Dhaliwal, N. A., Patil, I., & Cushman, F. (2021). Reputational and cooperative benefits of third-party compensation. *Organizational Behavior and Human Decision Processes*, 164, 27-51.
- Raihani, N. J., & Bshary, R. (2015). Third- party punishers are rewarded, but third- party helpers even more so. *Evolution*, 69(4), 993-1003.
- Reb, J., Goldman, B. M., Kray, L. J., & Cropanzano, R. (2006). Different wrongs, different remedies? Reactions to organizational remedies after procedural and interactional injustice. *Personnel Psychology*, 59(1), 31-64.

Referee: 2

Comments to the Author(s)

The current study examined the reputational benefits of third-party punishment (vs. third-party helping). The study was well-designed with clear research questions. I especially like that the fee-to-fine ratio was 1:1 in punishment, which allowed the authors to test the "punishment as a cooperative intent signal" more strictly. I have several concerns that need to be addressed.

- It was nice that the current findings provide converging evidence that helping is preferred over punishment in general (e.g., Jordan et al., 2016, Dhaliwal et al., 2021). However, it was unclear what the unique theoretical contribution of the current study is. Maybe authors can write a little bit more about how the current study complements the existing literature in the Introduction more. Also, I think the authors should cite the paper below which is very relevant to the current study:

Dhaliwal, N. A.*, Patil, I.*, & Cushman, F. A. (2021). Reputational and cooperative benefits of third-party compensation. *Organizational Behavior and Human Decision Processes*, 164, 27-51.

Thanks for this comment. We believe the main contribution of our study is to test the idea that investments in costly signals of punishment will escalate when there is the possibility to be chosen by a partner for a subsequent interaction. Based on previous studies of costly signalling, we might expect to observe this effect. And yet, because investing in punishment is an ambiguous signal of cooperative intent, there may be limits on the extent to which individuals use it to signal to others. Our data support this latter interpretation – as well as also calling into question the results of previous studies showing that investments in costly *helping* behaviour escalate when there is the possibility to be chosen as a partner. We have (hopefully) made this clearer in the substantially revised introduction and discussion sections of the paper (full text of changes appended to reviewer 1 comments and not re-pasted here to avoid repetition).

- I think the authors should explain more clearly what they manipulated in "Random vs. Partner Choice" and "Knowledge vs. No knowledge". It was unclear to what extent the third party was aware of the experimental setting. In the Knowledge condition, were third parties aware of the fact that bystanders will be informed of how the third parties behaved in the Help/Punishment stage?

Also, were third parties aware of the fact that the allocation will be random or partner choice? I thought that the Knowledge condition involved whether the bystander was informed of the third party's behavior (line #177), but was confused when it sounded like the condition was actually about the third party punisher's knowledge.

Thanks for these comments. Third-parties were explicitly informed about the potential consequences of their investments in Game A, specifically that they would need to be chosen or randomly selected to take part in Game B (in Partner Choice / Random Allocation conditions, respectively) and that the new partner would either know or not know how they had behaved in Game A (in the Knowledge / No Knowledge conditions, respectively). We have included the text from the experimental instructions (also presented in the SI) as follows:

Third-parties received different instructions about the potential consequences of their helping or punishment investment in Game B, according to the condition they were allocated to. Specifically, workers who were allocated to the Random Allocation condition saw text stating, “Game B. To take part in this game will have to be RANDOMLY selected to interact with a NEW worker”, whereas workers who were allocated to the Partner Choice condition saw text stating, “Game B. To take part in this game, you will have to be CHOSEN by a NEW worker, who will be Player 4.” Third-parties who were allocated to the No Knowledge condition also saw text stating, “Player 4 will NOT participate in GAME A and will NOT know how you behaved in GAME A.” Third-parties allocated to the Knowledge condition saw text stating, “Player 4 will NOT participate in GAME A but WILL know how you behaved in GAME A.” All experimental details, including instructions shown to participants, are available at { <https://osf.io/4zpkb/>}.

Please clarify your experimental manipulations. Also, it would be easier for readers to follow if authors explain expected results between conditions more clearly at the end of the Introduction section and explain the rationale why you varied Knowledge and Allocation so that readers can understand better what you tried to disentangle by manipulating these variables.

Thanks for the comment. We have included text at the end of the introduction as follows:

In addition to exploring whether individuals signal cooperative intent using punishment, we also aimed to replicate previous findings that costly investments in helpful behaviours also act as signals of cooperative intent and these investments escalate when there is the possibility to be chosen by a partner for a subsequent social interaction (Barclay and Barker, 2020; Barclay and Willer, 2007; Sylwester and Roberts, 2010). The prospect of positive reputation gains may increase investments in punishment (or helping) either because the third-party knows that their past behaviour will be revealed to others (who may treat them differently on the basis of previous actions) or because the third-party anticipates gaining access to social partners on the basis of their investment. To distinguish between these two possibilities, we replicate Barclay & Willer’s (2007) design by including a ‘Knowledge

Only' condition, in addition to a 'Knowledge + Partner Choice' condition. This allows us to test whether any increased investment relative to baseline stems from the fact that one's behaviour will be advertised to a potential social partner and whether investments escalate even further when individuals can compete to be chosen as a partner for future social interactions. If third-parties' investments are used to compete to access new social partners, then we expect them to be higher in the Knowledge + Partner Choice condition than they are in the Knowledge Only condition.

We have also tried to explain the reasons for including a Knowledge Only and a Knowledge + Partner Choice condition by adding an additional hypothesis to the paper (H1: Observability will increase investments in both (a) third-party punishment and (b) helpful behaviour relative to a baseline where no observation is possible.) Please see revised text below.

To summarize, in the current study we aim to extend the current signalling account of peer-punishment by testing whether:

- H1: observability will increase investments in both (a) third-party punishment and (b) helpful behaviour relative to a baseline where no observation is possible;
- H2: the possibility for partner choice leads to higher investments in (a) third-party punishment and (b) costly helpful actions compared to the mere observability of those actions by bystanders;
- H3: investments in (a) third-party punishment and (b) help positively predict probability to be chosen as a social partner by bystanders;
- H4: third-parties who invest more in (a) punishment and (b) help are trusted more by bystanders;
- H5: third-party investments in (a) punishment and (b) help are positively associated with trustworthiness.

- I have several clarifying questions about the data analyses. First, why did the authors convert a continuous dependent variable into a categorical one when testing H1, 2, and 4? I think the continuous DV is more fine-grained, and I couldn't find a clear reason for the decision. Could authors include the analyses with continuous DV in Supplementary Material? And/or explain more clearly why you had to turn continuous measure into a categorical one with arbitrary criteria?

- Second, I appreciate that the authors reported various analyses with the full and reduced sample. However, as a reader, it is very confusing to learn different results from Bayesian vs. frequentist and full vs. reduced sample. If possible, could the authors focus on one method (Bayesian vs. frequentist) and one sample (full vs. reduced) with clear rationale in the main text and move the rest to SOM?

We agree with the reviewer that it was potentially confusing to present two sets of analyses. Given that the main story of the paper is not sensitive to the analytic decision made, we opt to use the more widely-understood frequentist analyses in the paper and have removed all the Bayesian analyses from the manuscript. We hope this has simplified the take-home results. We have also made our analyses more internally consistent by analysing the data for Hypotheses H1 and H2 as a combined dataset, rather than splitting into punishment and helping conditions and analysing those separately. This does not affect the main take-home messages of the paper.

We hope that the results are easier to follow now that we have stuck to one approach throughout.

Relatedly, it is very confusing in the Results section (especially those from H1) because the results are slightly different across analyses. Could the authors end the results section (e.g., at the end of H1 results) with a summary sentence where you highlight the common patterns across different analyses, so that readers can digest the results more easily?

Please see comments above. We also included some summary statements about whether the hypotheses were supported or not.

- Third, I don't understand the sentence here "In the punishment condition, 44 participants (22 pairs) pairs invested the same amount in punishment and could not be used in this analysis." on page 13. Why can't they be analyzed? Could you elaborate on it? Further, I was confused why you used binomial tests here. Wouldn't it be more straightforward to run some sort of correlational test or GLM (e.g., probability of being chosen as a partner ~ invest.amount*condition) in which you don't have to throw away a part of your data.

This is a good question. We were specifically interested in whether bystander would choose the *higher* investor of the pair, so analysing the data in the way suggested doesn't necessarily make sense since an investment of e.g. 0.15 cents could be the higher investment in one pair but not in another. Therefore it is not invest amount specifically that we think ought to predict partner choice decisions but whether the amount invested was higher than what the partner invested.

Nevertheless, we would like to reassure the reviewer by additionally performing the logistic regression suggested, including all responses and not restricting to trials where both third-parties invested the same amount. We paste the methods and results of this analysis below; both are included in the main text of the paper.

Method:

H3: Investments in (a) third-party punishment and (b) help positively predict probability to be chosen as a social partner by bystanders

We addressed this hypothesis by performing two binomial tests asking whether the higher investor was more likely to be chosen in the help and punishment conditions, respectively. We also ran a chi-squared test to explore whether the probability of the higher investor being chosen varied according to condition (punishment / help). Because this approach meant that we could not use data from the 32 pairs (22 in punishment condition, 10 in help condition) where both third-parties invested the same amount, we also ran an additional model that allowed us to use all the data, including cases where both third-parties invested the same amount. Chosen (1/0) was set as the binomial response term in a logistic regression with the terms 'condition' (help / punishment) and 'Highest investor' (an ordered categorical variable with three levels, where lower < equal < higher denoting whether the third-party was the lower, equal or higher investor of the pair) set as explanatory terms. We also included the two-way interaction between these terms to estimate whether being the highest or lowest investor differentially impact the probability to be chosen in the help rather than the punishment condition.

Results:

H3: Investments in (a) third-party punishment and (b) help positively predict probability to be chosen as a social partner by bystanders

In the punishment condition, and in those groups where the two third-parties had invested different amounts, the highest investing player was chosen as the partner on 36 / 56 (64.3 %) occasions, indicating that bystanders used this information to select social partners (Binomial test, $p = 0.04$). In the help condition, and in those groups where the two third-parties had invested different amounts, the most helpful individual was chosen on 63/65 occasions (96.9 %, Binomial test, $p < 0.001$). The higher investor was more likely to be chosen in the help condition than in the punishment condition (Chi squared test, $\chi^2 = 19.4$, $p < 0.001$).

In the full model, including data from trials where both third-parties invested the same amount, the results were similar. Specifically, the effect of the lower or higher investor of the pair being chosen varied with condition. Lowest investors were less likely to be

chosen in the help condition compared to the punishment condition (estimate: 3.68, $p < 0.001$; Figure 2); and highest investors were more strongly preferred in the help condition than in the punishment condition (estimate: -2.88, $p < 0.001$; Figure 2).

- Lastly, could you explain your prior distributions of Bayesian analyses? From my understanding, the results can be different depending on which prior distribution you set for your model. Or are your models robust against different prior distributions?

This comment is no longer relevant because we have changed all analyses to be frequentist.

We hope we have satisfied all editorial and reviewer concerns about this manuscript, and we would like to thank the editor and reviewers for their extremely valuable feedback.

Best wishes on behalf of all authors

Nichola Raihani

Appendix B

The reviewers and I both agree that this is a responsive review that addresses most of the points raised in the prior review. R1 one has a clarification point on methods that should be addressed. My primary feedback, in line with points raised by both R1 and R2, concerns the intellectual framing of the paper and discussion. For example, R2 mentions the importance of articulating the implications of the results for the evolution of third-party punishment via indirect reciprocity, and R1 notes that the discussion is currently fairly descriptive rather than theory-driven. I agree with these points. For example, not explicitly addressing theoretical ideas concerning whether punishment versus help are better signals of cooperative intent in the introduction (the response letter mentions that this is because this is addressed in other papers) leaves the paper reading as more narrow and less relevant to a broader biological audience. It also means that some aspects of the paper, especially those looking at whether costly helping is treated as a signal of cooperative intent (introduced on pages 4-5), are motivated more in terms of methodological rather than theoretical questions. Given that the direct contrast between the efficacy of punishment and helping as a signal is a key part of the study design and results, the theoretical reason this is interesting needs to be made clear in both the introduction and results.

Thanks for the comment. We would like to thank you for giving us the opportunity to resubmit this manuscript as we are aware that Proc B does not allow multiple rounds of revision. We apologise for not sufficiently addressing these points in the previous revision and hope that you will find the current version much improved. We have replied to specific comments below in bold but summarise the main changes briefly here as well. In particular, we have (i) reframed the hypotheses, so that it is clear that we are investigating the signalling function of punishment *and* helping behaviour, as well as any differences between these; (ii) revised the presentation of the analysis and results section to be in keeping with the new hypotheses (no new analyses have been performed and no results have changed); and (iii) we have completely overhauled the discussion in light of the comments made by the editor and both referees. We have not made such substantive changes to the introduction because we already took a lot of space to outline why punishment might be a more ambiguous signal of cooperative intent than helping behaviour in this section (lines 76 to 127). We paste this below, also including a new section (highlighted) that we have included:

Nevertheless, the motives underpinning punishment decisions are hard to discern because punishment is, by definition, a harmful act. Punitive strategies could stem from spiteful motives, aimed at harming other people, or from fairness concerns and desires to uphold social norms of behaviour (Dhaliwal et al., 2020; Raihani and Bshary, 2019). Punishment is, by definition, a more ambiguous signal of cooperative intent than is helping, since helping others is not consistent with harmful or spiteful motives. Indeed, recent evidence shows that people who invest in third-party punishment (rather than third-party compensation) are more likely to score highly for antisocial personality traits, such as Machiavellianism, narcissism and psychopathy (Dhaliwal et al., 2021), suggesting that third-party punishment is less likely to be an honest signal of cooperative intent.

The information punishment conveys – and its associated reputation consequences – are likely to be highly context-specific (Raihani and Bshary, 2015a). For example, when individuals are given an option to compensate a victim or to punish a cheat, then punishment is less likely to signal cooperative intent (Dhaliwal et al., 2021; Heffner and FeldmanHall, 2019; Jordan et al., 2016; Raihani and Bshary, 2015b). Contexts

where the punisher is harmed directly by the wrongdoer ('second-party punishment') are more likely to be motivated by vengeful sentiments and less likely to be interpreted as signals of the punisher's cooperativeness (Heffner and FeldmanHall, 2019; Raihani and Bshary, 2015a). Conversely, when the punisher is an uninvolved bystander to the initial 'crime' ('third-party punishment'), then punitive acts are more likely to convey cooperative intent. Punishment that is cheaper to administer than the damage it inflicts on a target could also be perceived as a competitive act (Raihani and Bshary, 2019, 2015a), rather than signalling an intent to cooperate. The potential ambiguity and context-dependency of punishment means that, especially in decontextualised settings typical of economic games, the reputation consequences of punishing others might not always be positive (Dhaliwal et al., 2021; Heffner and FeldmanHall, 2019; Horita, 2010; Ozono and Watabe, 2012; Przepiorka and Liebe, 2016; Raihani and Bshary, 2015a). In some cases, therefore, individuals might choose to hide investments in punishing others (Rockenbach and Milinski, 2011) or preferentially invest in helping rather than punishing others when these decisions will be revealed to others (Li et al., 2021). Conversely, because helpful acts entail paying an individual cost to generate benefits to others, the motives underpinning these acts are less likely to be affected by context and may be more likely to be perceived as stemming from genuinely helpful motives (but see Raihani & Power 2021).

A simple hypothesis that does not take ambiguity and context-specificity into account might posit that if punishers stand to gain reputation benefits from punishing, then they should invest more in punishment when these decisions will be made known to other individuals. Similarly, we might expect punishers to further escalate their investments when there is competition to be selected as a social partner, as it has been previously shown for helping behaviour (Willer & Barclay, 2007). This hypothesis might, however, be unsupported if punishment is an ambiguous signal of cooperative intent and if increasing investments in punishment cast further doubt on the punisher's underlying intentions and commitment to cooperation. **Specifically, because the motives underpinning punishment may not indicate a cooperative disposition, we might not expect people to form favourable impressions of those who make escalated investments in punishment. As a consequence, people may not reward or trust or prefer to interact with people who invest high amounts in punishment, even if they do preferentially reward, trust and prefer to interact with people who invest high amounts in helpful behaviour.**

We have slightly amended the part of the introduction that gives the impression that the helping condition in our study was motivated by methodological rather than theoretical concerns, as follows (highlighted part is new):

In this study, therefore, we asked whether punishers do escalate their investments in punishment when there is the potential to be chosen by an observer – and whether punishers are preferred as interaction partners and entrusted with more money. We also examine whether investments in punishment act as honest signals of cooperative intent, by measuring whether punitive investments are related to trustworthy behaviour in a game where individuals are financially incentivised to exploit a partner. **Importantly, we compare whether investments in punishment escalate to the same extent as investments in helping behaviour – and whether these signals are treated differently by potential interaction partners. This part of the study aims to replicate**

previous results and to explore any differences in how people use and respond to signals of punishment versus signals of helping behaviour.

Finally, we have changed the structure of the hypotheses and also re-ordered the analyses and results to reflect this. Specifically, we have added two new hypotheses which pertain to the contrast between (i) investments in punishment and helping; (ii) whether punishers or helpers are preferred as interaction partner and (iii) whether punishment or helping is a more reliable signal of trustworthiness. Please see below:

To summarize, in the current study we tested the following hypotheses. H1 to H3 concern third-party signalling. H4-H6 concern whether bystanders selected third-parties on the basis of these signals; H7 and H8 deal with the extent to which third-party investments predict those individuals' trustworthiness.

Third-party signalling

- H1: Observability will increase investments in both (a) third-party punishment and (b) helpful behaviour relative to a baseline where no observation is possible.
- H2: The possibility for partner choice leads to higher investments in (a) third-party punishment and (b) costly helpful actions compared to the mere observability of those actions by bystanders.
- H3: Third-parties prefer to invest in helping (rather than punishment) when signalling to potential interaction partners.

Bystander choices

- H4: Investments in (a) third-party punishment and (b) help positively predict probability to be chosen as a social partner by bystanders.
- H5: Third-parties who invest more in (a) punishment and (b) help are trusted more by bystanders.
- H6: Bystanders place more weight on signals of help than on signals of punishment when (a) choosing and (b) trusting a partner for a cooperative interaction.

Reliability of third-party signals

- H7: Third-party investments in (a) punishment and (b) help are positively associated with trustworthiness.
- H8: Investments in third-party help more reliably predict trustworthiness than investments in third-party punishment.

We have also substantially overhauled the discussion and rather than pasting the entire discussion here. To save on space, we do not paste the new discussion here but we hope

this has addressed the comments about emphasizing the theoretical significance of the work and especially the implications for the evolution of punishment.

Reviewer(s)' Comments to Author:

Referee: 2

Comments to the Author(s)

I thank the authors for their reply to my comments and concerns. I believe their manuscript is much improved and clear now. I just have a few suggestions for final edits.

Thanks so much for the positive assessment.

I think the Discussion in the current manuscript is very descriptive (like a summary of results rather than a discussion of theoretical implications). It would be helpful for readers to highlight some theoretical implications of this work. One potential way to do this is by connecting the current findings with some hypotheses mentioned in the Intro. Personally, I'm interested in hearing about how the current findings inform the hypotheses in the Introduction e.g., "An alternative route by which punitive strategies could yield individual-level benefits to punishers is via reputation consequences that increase the punisher's likelihood to have profitable social interactions in the future: (i) by signalling formidability or (ii) by signalling cooperative intent." (line 45 on p. 2).

Thanks for the comment. =Please see the response to the editor above, where we indicate that we have completely overhauled the discussion in light of these comments. We hope that this now fully addresses these points.

Just to clarify, in the Helping condition, third parties were able to choose between helping a receiver and doing nothing (i.e., no investment). Similarly, in the Punishment condition, they chose between punishing a dictator and doing nothing (i.e., no investment). Is this correct? If so, some descriptions in the Method may be confusing to readers as it sounded like participants chose between helping and punishment, which was actually a between-subject variable. For example, from line 193 on p. 6, the authors wrote "Following the dictator decision, the third-party chose how much of their endowment, if any, to invest to punish an unfair Dictator or, according to the experimental condition, to help a receiver who was given an unfair share. Third-parties were endowed with \$0.50 and could invest any amount (in \$0.01 increments) between \$0.00 and \$0.45 to either help or punish the target individual, with a fee to fine/fee to help ratio of 1:1". It would be helpful for readers to be clear about the fact that there was a "no investment" option in both conditions, so the readers don't misunderstand helping vs. punishment as a within-subject variable.

Thanks for pointing this out – we have changed this wording so it is less confusing.

The authors called the current finding a "failed replication" (line 439 on p.15), which has a negative connotation. I would be more cautious about viewing this study as a failed replication because as the authors noted, there were differences in study designs between the two studies. So replacing "this failed replication" with "differences in results" or something similar would be more appropriate in this context.

Ok, we have done this.

Referee: 1

Comments to the Author(s)

The manuscript is much improved. That said, I think it would be helpful and interesting to the readers of this paper if the authors could tie everything together by explaining in a bit more detail what this collection of results might imply for the evolution of third-party punishment via indirect reciprocity.

Thanks for the positive assessment. We have completely overhauled the discussion, in line with comments from the editor and both referees.